# LANGUAGE MODEL BEATS DIFFUSION
# — TOKENIZER IS KEY TO VISUAL GENERATION

**Lijun Yu**[‡†*] **José Lezama**[†]   **Nitesh B. Gundavarapu**[†]   **Luca Versari**[†]   **Kihyuk Sohn**[†]

**David Minnen**[†] **Yong Cheng**[†] **Agrim Gupta**[†] **Xiuye Gu**[†] **Alexander G. Hauptmann**[‡]

**Boqing Gong**[†]    **Ming-Hsuan Yang**[†]    **Irfan Essa**[†]    **David A. Ross**[†]    **Lu Jiang**[†‡]

[†]Google, [‡]Carnegie Mellon University

## ABSTRACT

While Large Language Models (LLMs) are the dominant models for generative tasks in language, they do not perform as well as diffusion models on image and video generation. To effectively use LLMs for visual generation, one crucial component is the visual tokenizer that maps pixel-space inputs to discrete tokens appropriate for LLM learning. In this paper, we introduce MAGVIT-v2, a video tokenizer designed to generate concise and expressive tokens for both videos and images using a common token vocabulary. Equipped with this new tokenizer, we show that LLMs outperform diffusion models on standard image and video generation benchmarks including ImageNet and Kinetics. In addition, we demonstrate that our tokenizer surpasses the previously top-performing video tokenizer on two more tasks: (1) video compression comparable to the next-generation video codec (VVC) according to human evaluations, and (2) learning effective representations for action recognition tasks.

## 1   INTRODUCTION

Large transformer-based language models, commonly referred to as LMs or LLMs, are the de facto models for natural language generation (OpenAI, 2023; Google, 2023; Touvron et al., 2023). Over time, LMs have expanded their capabilities to generate content in various modalities, asserting their dominance in other domains like audio (Agostinelli et al., 2023), speech (Rubenstein et al., 2023), code generation (Li et al., 2023), medical applications (Singhal et al., 2023) and robotics (Zitkovich et al., 2023).

LMs are capable of generating images and videos. To do so, the image pixels are mapped into a sequence of discrete tokens by a visual tokenizer ($c.f.$ Section 2). These tokens are then fed into the LM transformer, as if they were lexical words, for generative modeling. Despite notable advancements in employing LMs for visual generation (Esser et al., 2021; Chang et al., 2022), LMs still do not perform as well as diffusion models (Rombach et al., 2022). For instance, when evaluating on the ImageNet dataset, a gold standard benchmark for image generation, the best language model (Lee et al., 2022) underperforms the diffusion model (Gao et al., 2023) by a substantial 48% margin (FID 3.41 $vs$. 1.79 when generating images at the $256 \times 256$ resolution).

*Why do language models lag behind diffusion models in visual generation?* This paper suggests that a primary reason is the lack of a good visual representation, resembling our natural language system, for effectively modeling the visual world. To substantiate this hypothesis, this paper shows that, when utilizing a good visual tokenizer, the masked language model (Devlin et al., 2019; Chang et al., 2022; Yu et al., 2023a) surpasses the state-of-the-art diffusion models in terms of both generation fidelity and efficiency across image and video benchmarks, given the same training data, comparable model size, and training budget. To the best of our knowledge, this provides the first evidence that language models beat diffusion models on the hallmark ImageNet benchmark.

It is worth emphasizing that our intention is not to assert whether the language model is superior to others, but to promote the exploration of visual tokenization methods for LLMs. A fundamental difference of LLMs from other models, such as diffusion models, is that LLMs utilize a discrete latent format: tokens obtained from a visual tokenizer. We show that the values of these discrete visual tokens should not be overlooked considering their distinct advantages as follows. **(1) Compatibility with LLMs.** The main advantage of a token representation is that it shares the same form

---

*Work done during a research internship at Google Research.

as language tokens, making it straightforward to leverage the optimizations our community has developed over many years for LLMs. This includes faster training and inference speeds (Shazeer, 2019; Lester et al., 2021), advancements in model infrastructure (Dao et al., 2022; Du et al., 2022), learning recipes for model scaling (Brown et al., 2020; Chowdhery et al., 2022), and GPU/TPU optimization, among other innovations. Unifying vision and language by the same token space could set the stage for a true multimodal LLM that can understand, generate, and reason within our visual environment. **(2) Compressed representation.** The discrete token may offer a fresh perspective on video compression. The visual tokens can serve as a new video compression format to reduce disk storage and bandwidth during internet transfers. Unlike compressed RGB pixels, these tokens can be fed directly into generative models, bypassing the conventional decompression and latent encoding steps. This allows for faster processing in generative video applications, especially beneficial in edge computing cases. **(3) Visual understanding benefits**. Prior research has shown that the discrete tokens are valuable as a pre-training target in self-supervised representation learning, as discussed in BEiT (Bao et al., 2021) and BEVT (Wang et al., 2022). Additionally, research finds that using tokens as the model inputs improves the robustness and generalization (Mao et al., 2021).

In this paper, we introduce MAGVIT-v2, a video tokenizer designed to map videos (and images) into compact discrete tokens. Our model is built on the state-of-the-art video tokenizer, MAGVIT (Yu et al., 2023a), within the VQ-VAE framework (Van Den Oord et al., 2017). We propose two new techniques. First, a novel lookup-free quantization method enables the learning of a large vocabulary that is able to improve generation quality of the language model. Second, through extensive empirical analyses, we have identified modifications to the tokenizer that not only enhance generation quality but also enable the tokenization of both images and videos using a shared vocabulary.

We empirically demonstrate that our model outperforms the previously top-performing video tokenizer, MAGVIT, in three key areas. First, our model significantly improves the generation quality of MAGVIT, establishing the state of the art on the common image and video benchmarks. Second, user studies indicate that its compression quality exceeds that of MAGVIT and the current video compression standard, HEVC (Sullivan et al., 2012). Moreover, it is on par with the next-generation video codec, VVC (Bross et al., 2021). Finally, we show that, compared to MAGVIT, our new tokens are stronger for video understanding tasks across two setups and three datasets. The main contributions of this work are:

- A new video tokenizer that outperforms the previously best-performing video tokenizer in three areas: visual generation, video compression, and action recognition.
- A novel lookup-free quantization approach that enables improving the visual generation quality of language models by learning a large vocabulary.
- To the best of our knowledge, the first evidence suggesting that a language model can outperform diffusion models on ImageNet when provided with the same training data, an equivalent model size, and a similar training budget.
- A video compressor with better quality than HEVC and VVC, at similar bit rates, according to user studies. To our knowledge, this is the first successful attempt of a visual tokenizer designed for video generation to achieve comparable results to standard codecs.

## 2 BACKGROUND

**Language Model (LM) for visual generation.** LMs have been extended to generate images and videos. A visual tokenizer $f$ is used to first map visual inputs into a sequence of discrete tokens. A video $\mathbf{V} \in \mathbb{R}^{T \times H \times W \times 3}$ (or image when $T = 1$) is tokenized into a discrete representation $\mathbf{X} = f(\mathbf{V}) \in \{1, 2, \cdots, K\}^{T' \times H' \times W'}$, where $K$ is the codebook (vocabulary) size of the visual tokenizer. $\mathbf{X}$ is flattened into a 1D token sequence obtained using raster scan ordering and then fed into an LM transformer for generative modeling.

Two types of LMs are commonly used for visual generation. The *Autoregressive LM (AR-LM)* includes ImageGPT (Chen et al., 2020), DALL-E (Ramesh et al., 2021), Parti (Yu et al., 2022b), *etc*. An AR-LM predicts the next token given the previous tokens along with additional conditioning information $\mathbf{c}$ using a categorical distribution for $p_\theta(\mathbf{x}_i \mid \mathbf{x}_{<i}; \mathbf{c})$. During inference, AR-LMs use the standard autoregressive decoding over the tokens. Finally, the tokens are converted back to pixels by a decoder associated with the visual tokenizer.

The *Masked LM (MLM)* is another type of language model for visual generation, such as: MaskGIT (Chang et al., 2022), MAGVIT (Yu et al., 2023a), Phenaki (Villegas et al., 2022), and MUSE (Chang et al., 2023), among others. An MLM is trained using a masked token objective (De-

vlin et al., 2019), where some tokens in the sequence are randomly masked and need to be predicted given the observed tokens. Let $\mathbf{m} \in \{0,1\}^n$ be a random binary sequence where $\mathbf{m}^\top \mathbf{1} \in [0, n-1]$. The MLM learns $p_\theta(\mathbf{x}_i \mid \{\mathbf{x}_j : \mathbf{m}_j = 1, \forall j\}; \mathbf{c})$ for all $i$ where $\mathbf{m}_i = 0$. To generate a video or image during inference, the MLM uses the non-autoregressive decoding algorithms for images and videos (Chang et al., 2022; Yu et al., 2023a). The decoding starts with a fully masked sequence, which is iteratively filled by repeating two steps: (1) sample the whole sequence $\hat{\mathbf{x}}^{(t)}$ from $p_\theta$ given the non-masked tokens from the previous step, (2) re-mask the $\lfloor \lambda(t) \cdot n \rfloor$ tokens in $\hat{\mathbf{x}}^{(t)}$ with the lowest probability, following a decreasing masking ratio schedule $\lambda(t)$, according to timestamp $t$.

**Denoising Diffusion Models (DDM).** DDMs (Sohl-Dickstein et al., 2015; Song & Ermon, 2019) are regarded as the state-of-the-art in visual generation due to their high-quality image (Dhariwal & Nichol, 2021; Ho et al., 2022a) and video generation (Ho et al., 2022c). For instance, DDPM (Ho et al., 2020) learns a denoising process parameterized as conditional Gaussian distributions over image pixels. Recently, diffusion models and language models have displayed a significant overlap. Recent DDMs diffuse over latents rather than raw pixels. These latents are obtained using models similar to the visual tokenizer used by LMs. In fact, the very first latent in diffusion, proposed by Rombach et al. (2022), is derived from a visual tokenizer. Additionally, the diffusion model's architecture has been shifting from the U-Net to the transformer architecture (Peebles & Xie, 2022). Consequently, the boundaries between diffusion and language models in visual generation have become less distinct. Yet, a fundamental difference between DDMs and LMs lies in the latent format, *i.e.*, continuous *vs*. discrete. We have discussed the benefits of having discrete tokens in Section 1 and will show that the proposed tokenizer improves in these aspects.

**Visual tokenization.** Visual tokenization plays an essential role in mapping pixels into a discrete representation suitable for generative modeling. VQ-VAE (Van Den Oord et al., 2017) is a cornerstone work in image tokenization. A VQ-VAE model consists of a convolutional neural network (CNN) encoder, a vector-quantization (VQ) bottleneck, and a CNN decoder. Given a video $\mathbf{V} \in \mathbb{R}^{T \times H \times W \times 3}$, the VQ-VAE's encoder $E$ produces latent embeddings $\mathbf{Z} = E(\mathbf{V}) \in \mathbb{R}^{T' \times H' \times W' \times d}$. Each embedding vector $\mathbf{z} \in \mathbb{R}^d$ in $\mathbf{Z}$ is then passed through the vector quantizer $q$, which assigns it to the closest entry $\mathbf{c} \in \mathbb{R}^d$ in the learned codebook embedding $\mathbf{C} \in \mathbb{R}^{K \times d}$:

$$q(\mathbf{z}) = \mathbf{c}_i, \text{ where } i = \underset{j \in \{1,2,\cdots,K\}}{\arg\min} \|\mathbf{z} - \mathbf{c}_j\|_2. \tag{1}$$

To get discrete tokens, we drop the embedding dimension and represent $\mathbf{Z}$ by its indices $\mathbf{X} \in \{1, 2, \cdots, K\}^{T' \times H' \times W'}$. For decoding, embeddings of all image tokens are given as input to the decoder $D$ to reconstruct the input $\hat{\mathbf{V}} = D(\mathbf{Z})$. Following VQ-VAE, VQGAN (Esser et al., 2021) introduces an adversarial loss and feature-level perceptual losses to enhance the image quality.

Video tokenization is more challenging and VQGAN has been adapted to meet this purpose (Ge et al., 2022; Villegas et al., 2022; Yu et al., 2023a). The state of the art in video tokenization is MAGVIT (Yu et al., 2023a), which introduces a better 3D architecture, an inflation technique for initialization using image pre-training, and robust training losses. With MAGVIT, the LMs achieve leading generation quality across multiple video benchmarks. However, MAGVIT struggles to tokenize images and often results in noticeable flickering in longer videos.

## 3 METHOD

We introduce a new **video tokenizer** designed to map the spatial-temporal dynamics from a visual scene into compact discrete tokens suitable for language models. Compared with image generation, video generation still faces substantial challenges in generating consistent and realistic motion. We are interested in exploring the capabilities of language models in tackling this unsolved challenge. Therefore, this paper focuses on a video tokenizer that can effectively represent video for generative modeling. Our approach builds upon the state-of-the-art video tokenizer, MAGVIT, as detailed in Yu et al. (2023a). This section highlights two new designs: a lookup-free quantizer and a collection of enhancements to the tokenizer model.

### 3.1 LOOKUP-FREE QUANTIZER

Although the community has made great progress in developing VQ-VAEs, the relationship between improvements in the reconstruction quality and subsequent generation quality is still not well understood. A common misconception is that improving reconstruction equates to improving the generation of the language model. For example, enlarging the vocabulary can improve reconstruction quality. However, such improvement only extends to generation when the vocabulary size is small, and a very large vocabulary can actually hurt the performance of the language model.

As illustrated by the dashed curves in Fig. 1, the reconstruction FID, indicated by the right $y$-axis (where a lower value is better), improves as the vocabulary size (the $x$-axis) increases. The orange solid curve in Fig. 1 represents the LM's generation quality (the left $y$-axis). The generation FID initially improves but deteriorates for larger vocabulary. This may shed light on why the vocabulary size of most language models for visual generation is around 1-8k (Esser et al., 2021; Villegas et al., 2022), which is significantly smaller than the size of natural language vocabulary, *i.e.* over 200k.

A simple trick for training a larger codebook involves decreasing the code embedding dimension when increasing the vocabulary size (Yu et al., 2022a). This trick captures the intuition of limiting the representational capacity of individual tokens, which in turn facilitates learning over the distribution of a large vocabulary.

**Lookup-Free Quantization (LFQ).** Motivated by the above observation, we reduce the VQ-VAE codebook's embedding dimension to zero. Formally, the codebook $\mathbf{C} \in \mathbb{R}^{K \times d}$ is replaced with an integer set $\mathbb{C}$ where $|\mathbb{C}| = K$. Recall that in VQ-VAE models, the quantizer must look up all $K$ $d$-dimensional embeddings in the codebook, where $d$ is typically 256, when computing the closest codebook entry to the encoder output. This new design eliminates the need for such embedding lookup entirely hence

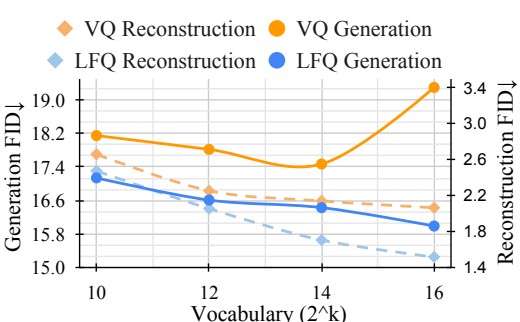

Figure 1: **Reconstruction and generation quality curves** in FID on ImageNet when scaling the tokenizer's vocabulary size with Vector Quantization (VQ) and Lookup-Free Quantization (LFQ). Comparison is done at $128 \times 128$ resolution using an MLM with 306-372M parameters.

we call it *lookup-free quantization (LFQ)*. We found that LFQ can grow the vocabulary size in a way benefiting the generation quality of language models. As shown by the blue curves in Fig. 1, both reconstruction and generation consistently improves as the vocabulary size increases – a property not observed in current VQ-VAE methods.

While various LFQ methods are available, this paper discusses a straightforward variant that assumes independent codebook dimensions and binary latents. Specifically, the latent space of LFQ is decomposed as the Cartesian product of single-dimensional variables, as $\mathbb{C} = \times_{i=1}^{\log_2 K} C_i$. Given a feature vector $\mathbf{z} \in \mathbb{R}^{\log_2 K}$, each dimension of the quantized representation $q(\mathbf{z})$ is obtained from:

$$q(\mathbf{z}_i) = C_{i,j}, \text{ where } j = \arg\min_k \|\mathbf{z}_i - C_{i,k}\|, \tag{2}$$

where $C_{i,j}$ is the $j$-th value in $C_i$. With $C_i = \{-1, 1\}$, the $\arg\min$ can be computed by the sign function as

$$q(\mathbf{z}_i) = \text{sign}(\mathbf{z}_i) = -\mathbb{1}\{\mathbf{z}_i \leqslant 0\} + \mathbb{1}\{\mathbf{z}_i > 0\}. \tag{3}$$

With LFQ, the token index for $q(\mathbf{z})$ is given by:

$$Index(\mathbf{z}) = \sum_{i=1}^{\log_2 K} \arg\min_k \|\mathbf{z}_i - C_{i,k}\| \prod_{b=0}^{i-1} |C_b| = \sum_{i=1}^{\log_2 K} 2^{i-1} \mathbb{1}\{\mathbf{z}_i > 0\}, \tag{4}$$

where $|C_0| = 1$ sets the virtual basis.

We add an entropy penalty during training to encourage codebook utilization:

$$\mathcal{L}_{entropy} = \mathbb{E}[H(q(\mathbf{z}))] - H[\mathbb{E}(q(\mathbf{z}))]. \tag{5}$$

This penalty is inspired by a similar loss used in image VQGAN model (Chang et al., 2022), which is also found in entropy-based clustering (Jansen et al., 2020). In LFQ, given the independence among dimensions, we rewrite $H(q(\mathbf{z})) = \sum_{i=1}^{\log_2 K} H(q(\mathbf{z}_i))$. The $H[\mathbb{E}(q(\mathbf{z}))]$ term can be approximated with sub-groups of dimensions for $K > 2^{18}$ where direct estimation is memory bound.

We note that there are various other variants of LFQ, *e.g.*, opting for the multivariant over the binary codebook $C_i$ or employing other quantization techniques such as Agustsson et al. (2019). As the first paper to introduce this concept, we focus on the simplest form with independent binary dimensions, which shows promising improvements. Other LFQ methods merit further research.

In addition to the entropy penalty (Eq. (5)), an LFQ-based tokenizer is trained using the standard combination of *reconstruction*, *GAN*, *perceptual*, and *commitment* losses (Esser et al., 2021), excluding the inapplicable codebook loss. Following Yu et al. (2023a), we use LeCAM regularization (Tseng et al., 2021) for improved stability.

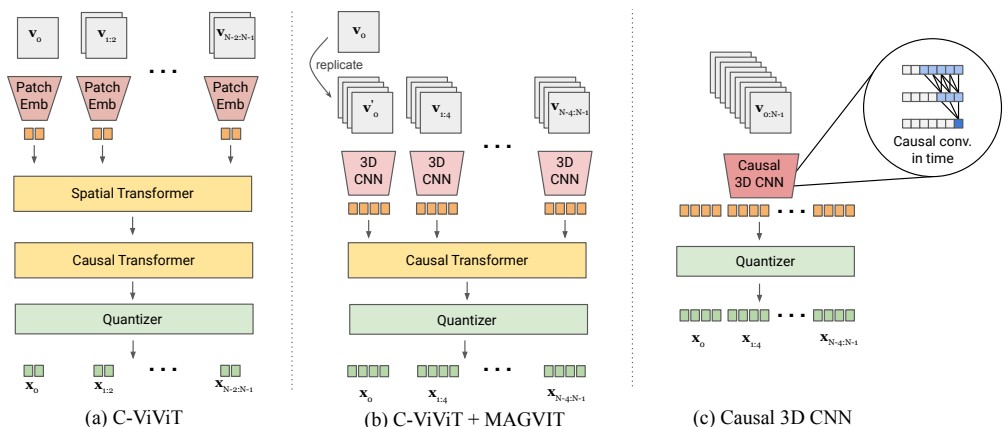

(a) C-ViViT       (b) C-ViViT + MAGVIT       (c) Causal 3D CNN

Figure 2: **Causal tokenizer architecture comparison**. The decoders, which are omitted from the figure, employ an architecture that is symmetric to the encoder. See detailed architecture diagram in the Appendix.

## 3.2 VISUAL TOKENIZER MODEL IMPROVEMENT

**Joint image-video tokenization.** A desirable feature of visual tokenization is the capability to tokenize images and videos using a shared codebook. However, the MAGVIT tokenizer, which utilizes the 3D CNN, faces challenges in tokenizing images due to the temporal receptive field.

To build a joint image-video tokenizer, a new design is needed. We begin our discussion by revisiting an existing method C-ViViT (Villegas et al., 2022). As depicted in Fig. 2a, C-ViViT employs full spatial transformer blocks combined with causal temporal transformer blocks. This approach performs reasonably well but has two drawbacks. First, unlike CNNs, the positional embeddings makes it difficult to tokenize spatial resolutions that were not seen during training. Second, empirically we found that 3D CNNs perform better than spatial transformer and produce tokens with better spatial causality of the corresponding patch.

To tackle these drawbacks, we explore two plausible designs. Fig. 2b combines C-ViViT and MAGVIT. Assuming a temporal compression ratio of 4, a 3D CNN processes blocks of 4 frames followed by a causal transformer. In Fig. 2c, we use the temporally causal 3D convolution to replace the regular 3D CNN. Specifically, the temporal padding scheme for a regular 3D convolution layer with kernel size $(k_t, k_h, k_w)$ includes $\lfloor \frac{k_t-1}{2} \rfloor$ frames before and $\lfloor \frac{k_t}{2} \rfloor$ frames after the input frames. In contrast, a causal 3D convolution layer pads with $k_t - 1$ frames before the input and nothing after, so that the output for each frame only depends on the previous frames. In consequence, the first frame is always independent of other frames, allowing the model to tokenize single images.

Temporal convolutional subsampling with stride $s$ is sufficient for $s\times$ down-sampling by mapping $1 + s \times t$ frames into $1 + t$. After a regular $s\times$ up-sampling, we drop the first $s - 1$ resulting frames, which maps $1 + t$ frames into $1 + s \times t$ and allows for the tokenization of a single image. Tab. 5a empirically compares the designs in Fig. 2, and we find that the causal 3D CNN performs the best.

**Architecture modifications.** In addition to using causal 3D CNN layers, we made several other architectural modifications to improve upon the MAGVIT model. First, we change the encoder downsamplers from average pooling into strided convolutions to leverage learned kernels, and replace the decoder upsamplers from nearest resizing followed by convolution with a depth-to-space operator. Second, we defer the temporal downsampling from the first few encoder blocks to the last ones. In addition, the downsampling layer in the discriminator now utilizes 3D blur pooling (Zhang, 2019) to encourage shift invariance. Finally, we add one adaptive group normalization layer before the residual blocks at each resolution in the decoder to pass in the quantized latents as the control signal following StyleGAN (Karras et al., 2019). Tabs. 5b and 5c empirically verify these designs.

**Token factorization for efficient prediction.** The output tokens can be fed into language models to generate videos. To assist smaller transformers predicting in a large vocabulary, we can factorize the LFQ token's latent space into equal subspaces. For instance, rather than predicting using a codebook of size $2^{18}$, we can predict in two concatenated codebooks, each of size $2^9$. We embed each subspace token separately and use their embedding summation as the token embedding for the transformer input. We find it beneficial to use weight tying (Press & Wolf, 2017), a common technique in language modeling, which involves sharing the weights between the embedding and softmax layers. For the output layer with a factorized vocabulary, we use the embedding matrix for each subspace to obtain logits with seperate prediction heads.

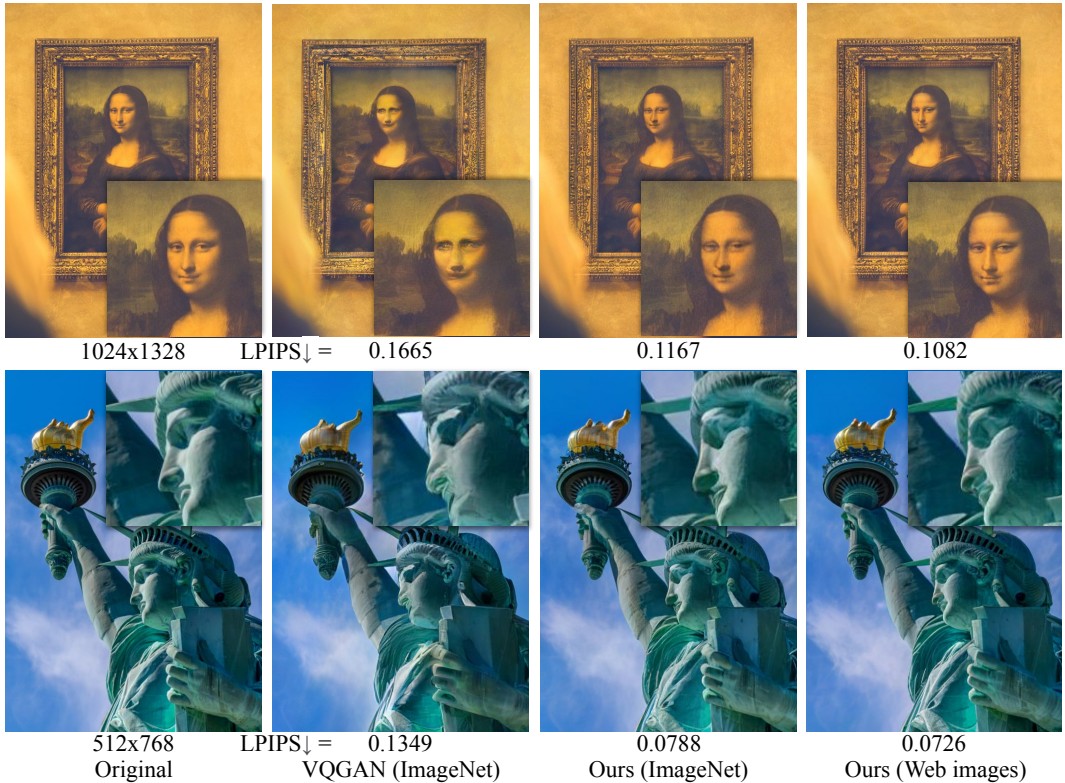

| 1024x1328 | LPIPS↓ = | 0.1665 | 0.1167 | 0.1082 |

| 512x768 | LPIPS↓ = | 0.1349 | 0.0788 | 0.0726 |
| Original | | VQGAN (ImageNet) | Ours (ImageNet) | Ours (Web images) |

Figure 3: **Image reconstruction samples with different tokenizers**. We compare the VQGAN used in MaskGIT (Chang et al., 2022) with two of our models trained on ImageNet and web images (Chen et al., 2022). Original images are by Eric TERRADE and Barth Bailey on Unsplash.

## 4 EXPERIMENTS

This section empirically verifies the proposed tokenizer across three distinct tasks: video and image generation, video compression, and action recognition. Fig. 3 visually compares the reconstruction quality of our tokenizer with prior works. More qualitative samples are shown at https://magvit.cs.cmu.edu/v2.

### 4.1 EXPERIMENTAL SETUPS

**Datasets.** We use Kinetics-600 (K600) (Carreira et al., 2018) and UCF-101 (Soomro et al., 2012) for video generation experiments, along with ImageNet (Deng et al., 2009) for image generaton. In addition, MCL-JCV (Wang et al., 2016) is used as the testbed for video compression, with Kinetics-400 (K400) (Kay et al., 2017) and SSv2 (Goyal et al., 2017) for video understanding.

**Implementation details** We follow the tokenizer training setting and hyperparameters in (Yu et al., 2023a), unless stated otherwise. LFQ is used, which eliminates the codebook embedding, to increase the default codebook size to $K = 2^{18}$. The weight of $\mathcal{L}_{entropy}$ follows an annealing schedule with a $3\times$ higher starting point and linearly decays to a fixed value of $0.1$ within 2k steps. We defer details regarding the evaluation setup of each subsection to the Appendix.

### 4.2 VISUAL GENERATION

The masked language model (MLM) (Devlin et al., 2019) is used in image and video generation. To verify the tokenizer, we employ the same MLM transformers in MAGVIT (Yu et al., 2023a). We select the MLM due to its competitive performance on benchmark datasets (Yu et al., 2023a; Lezama et al., 2023). In the Appendix, we also show that an autoregressive language model (AR-LM) coupled with the proposed tokenizer outperforms the prior work MAGVIT. As we use a smaller MLM (∼300M parameters) with a large codebook ($2^{18} \approx 262$K), the token factorization as discussed in Section 3.2 is applied using two heads with each predicting from a codebook of size $2^9$.

**Video generation.** We consider two standard video benchmarks, UCF-101 for class-conditional generation and K600 for frame prediction with 5-frame condition. FVD (Unterthiner et al., 2018) is used as our primary evaluation metric. Tab. 1 shows that our model surpasses all prior arts in both

Table 1: **Video generation results**: frame prediction on Kinetics-600 and class-conditional generation on UCF-101. We adopt the evaluation protocol of MAGVIT.

| Type | Method | K600 FVD↓ | UCF FVD↓ | #Params | #Steps |
|------|--------|-----------|----------|---------|--------|
| GAN | TrIVD-GAN-FP (Luc et al., 2020) | 25.7±0.7 | | | 1 |
| Diffusion | Video Diffusion (Ho et al., 2022c) | 16.2±0.3 | | 1.1B | 256 |
| Diffusion | RIN (Jabri et al., 2023) | 10.8 | | 411M | 1000 |
| AR-LM + VQ | TATS (Ge et al., 2022) | | 332±18 | 321M | 1024 |
| MLM + VQ | Phenaki (Villegas et al., 2022) | 36.4±0.2 | | 227M | 48 |
| MLM + VQ | MAGVIT (Yu et al., 2023a) | 9.9±0.3 | 76±2 | 306M | 12 |
| MLM + LFQ | non-causal baseline | 11.6±0.6 | | 307M | 12 |
| MLM + LFQ | *MAGVIT-v2 (this paper)* | 5.2±0.2
**4.3**±**0.1** | **58**±**3** | 307M | 12
24 |

Table 2: **Image generation results**: class-conditional generation on ImageNet 512×512. Guidance indicates the classifier-free diffusion guidance (Ho & Salimans, 2021). * indicates usage of extra training data. We adopt the evaluation protocol and implementation of ADM.

| Type | Method | w/o guidance | | w/ guidance | | #Params | #Steps |
|------|--------|------|------|------|------|---------|--------|
| | | FID↓ | IS↑ | FID↓ | IS↑ | | |
| GAN | StyleGAN-XL (Sauer et al., 2022) | | | 2.41 | 267.8 | 168M | 1 |
| Diff. + VAE* | DiT-XL/2 (Peebles & Xie, 2022) | 12.03 | 105.3 | 3.04 | 240.8 | 675M | 250 |
| Diffusion | ADM+Upsample (Dhariwal & Nichol, 2021) | 9.96 | 121.8 | 3.85 | 221.7 | 731M | 2000 |
| Diffusion | RIN (Jabri et al., 2023) | 3.95 | 216.0 | | | 320M | 1000 |
| Diffusion | simple diffusion (Hoogeboom et al., 2023) | 3.54 | 205.3 | 3.02 | 248.7 | 2B | 512 |
| Diffusion | VDM++ (Kingma & Gao, 2023) | 2.99 | 232.2 | 2.65 | 278.1 | 2B | 512 |
| MLM + VQ | MaskGIT (Chang et al., 2022) | 7.32 | 156.0 | | | 227M | 12 |
| MLM + VQ | DPC+Upsample (Lezama et al., 2023) | 3.62 | 249.4 | | | 619M | 72 |
| MLM + LFQ | *MAGVIT-v2 (this paper)* | 4.61
3.07 | 192.4
213.1 | **1.91** | **324.3** | 307M | 12
64 |

benchmarks. Specifically, it outperforms the previous best model MAGVIT by a large margin, while using the same MLM transformer backbone. In addition, it significantly outperforms the non-causal baseline on frame prediction, highlighting the contribution of the causal tokenizer. These results demonstrate the essential role of a good visual tokenizer in enabling LMs to generate high-quality videos. Fig. 4 shows qualitative samples from the model.

**Image generation on ImageNet.** We evaluate MAGVIT-v2 on image generation under the standard ImageNet class-conditional setting. We present results for resolution 512×512 in Tab. 2 and refer to the Appendix for 256×256 results. FID (Heusel et al., 2017) and Inception Score (IS) (Salimans et al., 2016) are used as evaluation metrics. Our model surpasses the best performing diffusion models both in sampling quality (FID and IS) and inference-time efficiency (sampling steps).

It is worth noting that all the models compared are trained using the same ImageNet training data, with a comparable model size and training budget. Therefore, the performance primarily evaluates the model's capabilities. The masked language model, equipped with our tokenizer, exhibits a notable improvement in FID over the best diffusion model baseline at 512×512 (FID=1.91 *vs.* 2.65, 28%↓). While this margin narrows at 256×256 resolution, the MLM uses a 50% reduced model size and needs much fewer decoding steps (*e.g.*, 64 *vs.* 250) to get the image generation quality. Qualitative samples in comparison with other models are shown in Fig. 5.

## 4.3 VIDEO COMPRESSION

We conduct a subjective rater study to assess the compression quality of MAGVIT-v2. The study is conducted on the 30 videos of the MCL-JCV dataset, resized to a resolution of 640×360. Sixteen raters are engaged, each providing responses to an average of roughly 800 pairwise-preference questions.

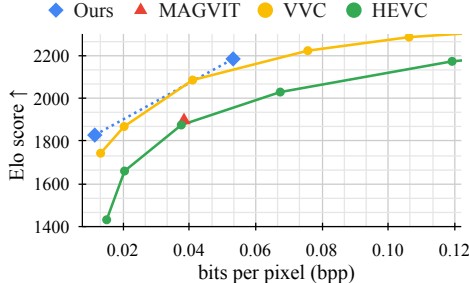

Figure 6: **Video compression rater study**.

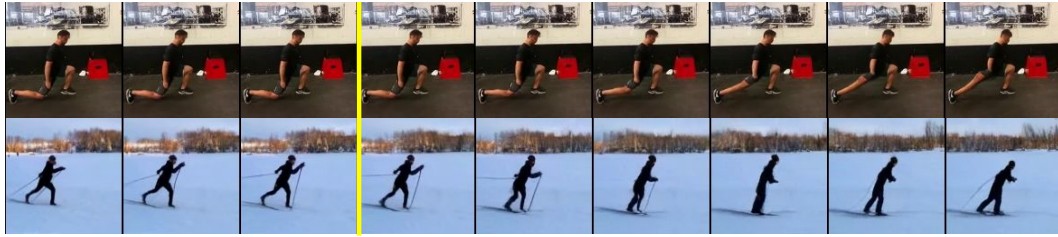

Condition → Generation

Figure 4: **Frame prediction samples on Kinetics-600**.

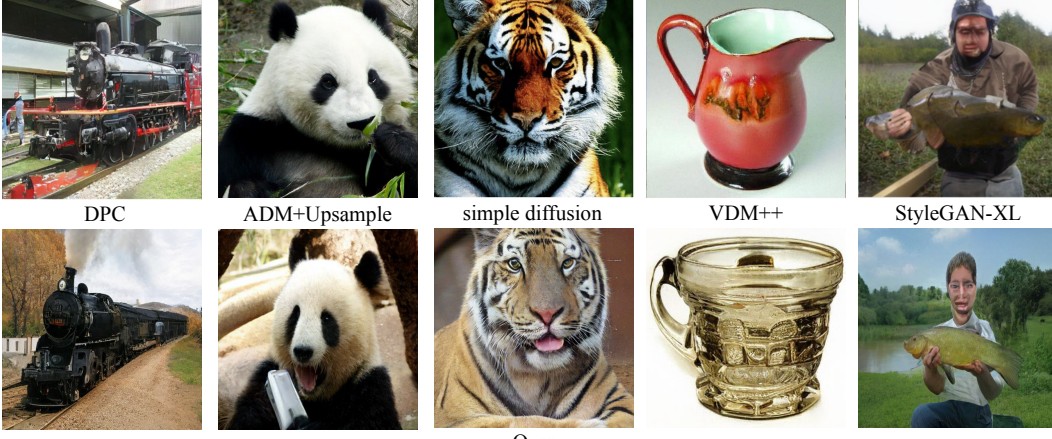

| DPC | ADM+Upsample | simple diffusion | VDM++ | StyleGAN-XL |

Ours

Figure 5: **Class-conditional generation samples on ImageNet 512×512**. We compare with each of the previous works with a random sample from the same image class.

We calculate Elo scores (Elo & Sloan, 2008) based on pairwise preferences to quantify the relative visual quality between the models. The study compares our model with MAGVIT as well as the current video compression standard HEVC (H.265) video codec (Sullivan et al., 2012) and the next-generation codec VVC (H.266) (Bross et al., 2021). As shown in Fig. 6, raters prefer our model to the compared methods at multiple bit rates.

We also compare the compression quality using common distortion metrics (LPIPS, PSNR, and MS-SSIM). Tab. 3 compares at 0.0384 bpp, the bit rate of MAGVIT, with full curves in the Appendix. The results show that our model outperforms MAGVIT on all metrics, and it outperforms all methods on

Table 3: **Video compression metrics**.

| Method | LPIPS↓ | PSNR↑ | MS-SSIM↑ |
|---|---|---|---|
| HEVC (Sullivan et al., 2012) | 0.199 | 30.10 | 0.943 |
| VVC (Bross et al., 2021) | 0.153 | **32.65** | **0.966** |
| MAGVIT (Yu et al., 2023a) | 0.144 | 23.70 | 0.846 |
| *MAGVIT-v2 (this paper)* | **0.104** | 26.18 | 0.894 |

LPIPS, a metric which correlates more closely with subjective quality assessments. At equal bit rates, standard codecs may render local details more accurately than neural models but also introduce block artifacts, detrimental to perceptual quality yet not captured by PSNR and MS-SSIM (Agustsson et al., 2019). Despite promising results with TPUs, further research is needed to adapt our model to run efficiently on CPUs like standard codecs.

## 4.4 VIDEO UNDERSTANDING

In this subsection, we assess the tokenizer's capability to learn a video understanding model for action recognition. Two setups are examined: (1) using tokens as prediction targets for the transformer's output, and (2) using tokens as the input to the transformer. For the former setup, we use a similar architecture following the BEVT (Wang et al., 2022) pre-training. For

Table 4: **Video action recognition performance** (classification accuracy↑ ×100).

| Token as transformer's: | Output | Input | | |
|---|---|---|---|---|
| Tokenizer | SSv2 | SSv2 | K400 | K600 |
| 3D VQ-VAE | 64.13 | 41.27 | 44.44 | 45.67 |
| MAGVIT (Yu et al., 2023a) | 67.22 | 57.34 | 72.29 | 74.65 |
| *MAGVIT-v2 (this paper)* | **67.38** | **62.40** | **75.34** | **77.93** |
| Raw pixel | 64.83 | 63.08 | 76.13 | 78.92 |
| HoG descriptor (Wei et al., 2022) | 65.86 | n/a | n/a | n/a |

the tokens as inputs, to work with the ViViT backbone (Arnab et al., 2021), we detokenize the tokens to pixels before feeding them to frozen ViViT transformers trained on raw pixels.

Tab. 4 shows that MAGVIT-v2 outperforms the previous best MAGVIT in these evaluations. Specifically, when using the decoded tokens as input, the performance approaches that of the model trained

Table 5: **Ablation study verifying key design choices**.

(a) Causal architectures on UCF-101. FID is calculated on the first frame.

| | #Params | FID↓ | FVD↓ |
|---|---|---|---|
| MAGVIT | 39M | n/a | 107.15 |
| C-ViViT | 90M | 28.02 | 437.54 |
| C-ViViT + MAGVIT | 67M | 13.52 | 316.70 |
| *MAGVIT-v2*: Causal 3D CNN | 58M | **7.06** | **96.33** |

(b) Image tokenization on ImageNet 128×128.

| | FID↓ | LPIPS↓ |
|---|---|---|
| MAGVIT | 2.65 | 0.1292 |
| + LFQ | 2.48 | 0.1182 |
| + large vocabulary | 1.34 | 0.0821 |
| + up/downsampler | 1.21 | 0.0790 |
| + deeper model | 1.20 | 0.0686 |
| + adaptive normalization | **1.15** | **0.0685** |

(c) Video tokenization on UCF-101.

| | FVD↓ | LPIPS↓ |
|---|---|---|
| MAGVIT | 24.55 | 0.0988 |
| + LFQ & large vocabulary | 16.12 | 0.0694 |
| + up/downsampler | 15.37 | 0.0678 |
| + late temporal downsample | 11.11 | 0.0653 |
| + deeper model | 8.90 | 0.0542 |
| + 3D blur pooling | **8.62** | **0.0537** |

with ground-truth pixels using the same ViViT backbone. While these numbers are still worse than the state-of-the-art in action recognition, they represent solid improvements credited to the new tokenizer.

## 4.5 ABLATION STUDY

In Fig. 1, we have ablated LFQ *vs*. VQ and the vocabulary size. In Tab. 5, we validate the key designs proposed in Section 3.2. Specifically, Tab. 5a compares the architecture illustrated in Fig. 2; Tab. 5b and Tab. 5c verify the LFQ and other improvements on ImageNet and UCF-101, respectively.

## 5 RELATED WORK

**Visual tokenization.** Beyond the VQ-VAE models discussed in Section 2, additional models have been proposed. ViT-VQGAN (Yu et al., 2022a) introduces transformer blocks as a substitute for CNNs for image tokenization. C-ViViT (Villegas et al., 2022) further extends this idea for video tokenization. Early studies on video tokenization treat frames as independent images with no temporal compression (Wu et al., 2022; Gupta et al., 2022). Later research (Yan et al., 2021; Ge et al., 2022; Yu et al., 2023a) integrates 3D CNNs to tokenize spatial-temporal volumes. Despite these advances in vector quantization (VQ), the codebook learned by previous VQ models is relatively small (*e.g.*, 8k) due to the difficulty in improving the generation quality with larger vocabularies. In contrast, our tokenizer can induce a large vocabulary (*e.g.*, 262k) that can be effectively modeled by an LM, leading to enhanced image and video generation quality.

**Text-to-{image, video}.** Text-to-image and text-to-video generation has garnered significant rapid advancements using both language models (Yu et al., 2023b; Chang et al., 2023) and diffusion models (Ho et al., 2022a; Blattmann et al., 2023; Singer et al., 2022; Ge et al., 2023; Ramesh et al., 2022). Although diffusion models, such as Midjourney, are considered the top performers in these tasks, it is unclear whether their advantage stems from the model, data, or some other unidentified factors. Indeed, it is challenging to scientifically compare these text-to-image models as they are trained on varied datasets, with some even being proprietary data, under inconsistent training conditions. To facilitate a fairer comparison, this paper prioritizes using the ImageNet and Kinetics benchmarks.

**Diffusion models.** Exhibiting high quality sampling, pixel-space diffusion models (Sohl-Dickstein et al., 2015; Song & Ermon, 2019; Ho et al., 2020) raised to the top of the generative modeling space for both image (Ho et al., 2020; Dhariwal & Nichol, 2021; Saharia et al., 2022) and video (Ho et al., 2022c;a; Singer et al., 2022) synthesis. The pixel-space denoising diffusion models (DDMs) are later refined by the latent-space DDM (Rombach et al., 2022), which conducts diffusion over the *continuous* latent embeddings derived from a pre-trained variational autoencoder (VAE). Binary latents for image modeling were used in Wang et al. (2023), where the diffusion process is parameterized with Bernoulli distributions. Recent studies have identified advantages in substituting the U-Net (Ronneberger et al., 2015) denoising backbone with a Transformer (Peebles & Xie, 2022; Jabri et al., 2023) or a hybrid of both (Hoogeboom et al., 2023), making the distinctions between diffusion and language models in visual generation more blurred, with a key distinction being their latent format — continuous for diffusion and discrete for language models.

## 6 CONCLUSION AND FUTURE WORK

We introduce MAGVIT-v2, a novel video tokenizer that exploits lookup-free quantization along with architectural advancements to tokenize images and video with a shared vocabulary. The experiments show that our tokenizer outperforms the previously leading video tokenizer across three areas: visual generation, video compression, and action recognition in videos. Our results suggest that a good visual tokenizer is key for enabling language models to excel in image and video generation. These results demonstrate the great capabilities of LMs in visual generation, and advocate for further exploration of advanced visual tokenization methods designed for LLMs.

ACKNOWLEDGMENTS

We would like to express our gratitude to Yu-Chuan Su and Sergey Ioffe for their valuable comments on our work, to Josh Dillon for discussions, and to Eirikur Agustsson for help in compression evaluation.

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

# A  IMPLEMENTATION DETAILS

## A.1  IMAGE AND VIDEO GENERATION

We set up two image tokenizers to downsample by $16\times$ and $32\times$, where they are used for generation at $256\times256$ and $512\times512$, respectively. In both cases, an image is represented as $16\times16$ tokens. We train them on the ImageNet training set for 270 epochs using a batch size of 256, both with $256\times256$ images.

With this tokenizer we train a Masked Language Model following Yu et al. (2023a), using the token factorization described in Section 3.2. We train for 1080 epochs in accordance with the prior best model MDT (Gao et al., 2023), with batch size 1024 for better efficiency. For preprocessing and data augmentation, we randomly crop 80-100% of an image while keeping the aspect ratio, followed by random horizontal flipping. The class label is dropped for 10% of the training batches to enable classifier-free guidance (Ho & Salimans, 2021). For unguided generation, we use temperature 30 for $512\times512$ and 15 for $256\times256$ in the non-autoregressive decoding. For guided generation, we adopt the guidance schedule from Gao et al. (2023) with temperature scaling (Lezama et al., 2023), where we use guidance scale 25 with temperature 15.

We inflate an image tokenizer trained at $128\times128$ for video modeling. Different from the inflation in Yu et al. (2023a), we fill in the temporally last slice to correspond to the causal padding scheme. In addition, we disable the inflation for the discriminator and train it from scratch for better stability. We train the causal video tokenizer on Kinetics-600 training set for 190 epochs with batch size 256. This tokenizer is also used in subsequent evaluations of video compression and action recognition.

With the causal tokenizer producing $5\times16\times16$ tokens for a $17\times128\times128$ clip, the first $2\times16\times16$ tokens are provided as the condition of the first 5 frames, per the standard setup of Kinetics-600 frame prediction benchmark. We train the MLM transformer following Yu et al. (2023a) with token factorization for 360 epochs with batch size 256. The model is sampled with a cosine schedule using temperature 32.

## A.2  MODEL SETUP AND HYPERPARAMETERS

Fig. 7 illustrates the architecture of our proposed MAGVIT-v2. We provide detailed training hyperparameters for our models as listed below:

- Video input: 17 frames, frame stride 1, $128 \times 128$ resolution.
- Base channels: 128.
- VQVAE channel multipliers: $1, 2, 2, 4$.
- Discriminator channel multipliers: $2, 4, 4, 4, 4$.
- Number of residual blocks: 4.
- Latent shape: $5 \times 16 \times 16$.
- Vocabulary size: $2^{18}$.
- Initialization: central inflation from a 2D model trained on ImageNet with this setup.
- Entropy loss weight: 0.1.
- Entropy loss annealing steps: 2000.
- Entropy loss annealing factor: 3.
- Reconstruction loss weight: 5.0.
- Generator loss type: Non-saturating.
- Generator adversarial loss weight: 0.1.
- Discriminator gradient penalty: r1 with cost 10.
- Perceptual loss weight: 0.1.
- Commitment loss weight: 0.25.
- LeCAM weight: 0.001.
- Peak learning rate: $10^{-4}$.
- Learning rate schedule: linear warm up and cosine decay.
- Optimizer: Adam with $\beta_1 = 0$ and $\beta_2 = 0.99$.
- EMA model decay rate: 0.999.
- Batch size: 256.

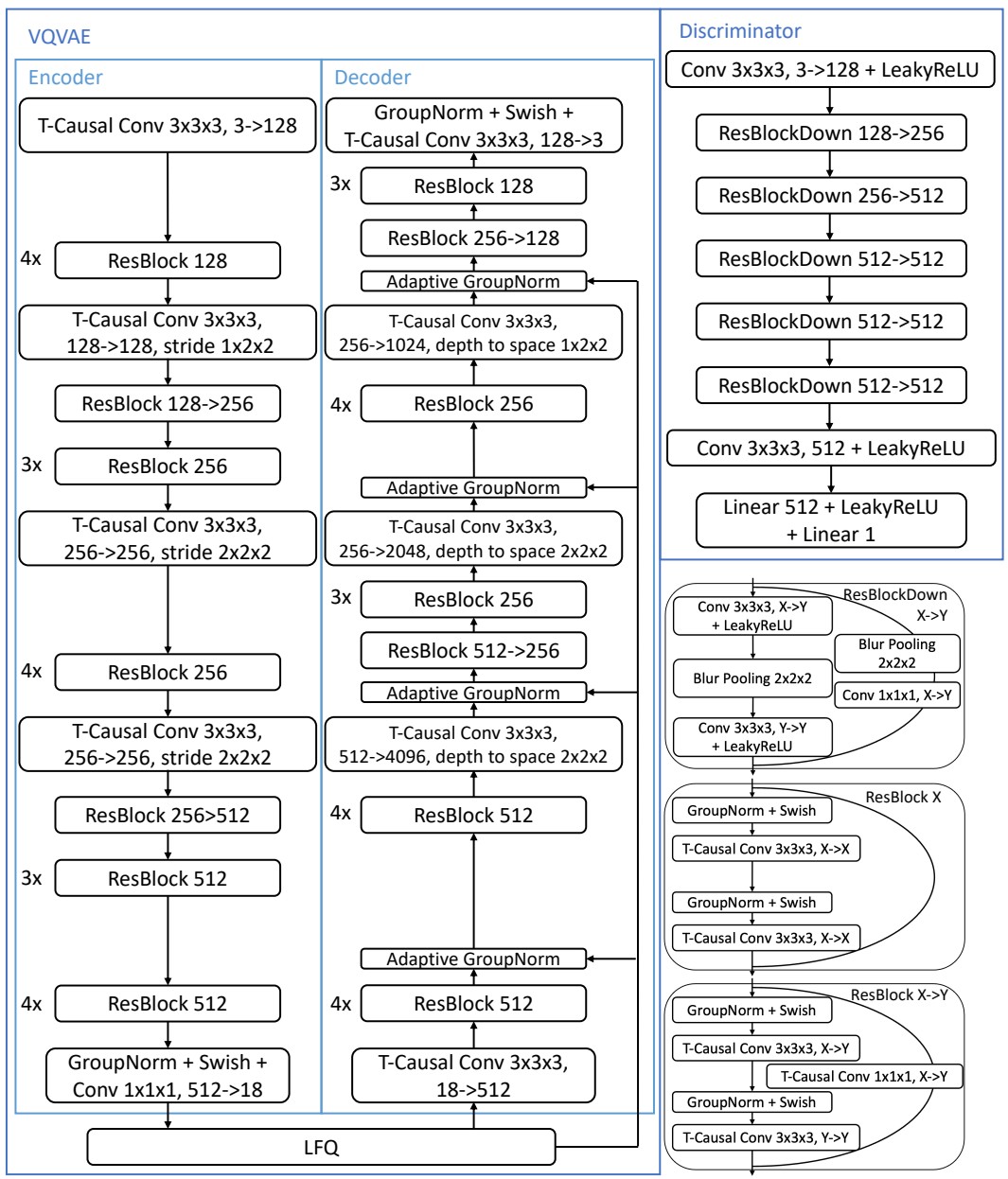

Figure 7: **MAGVIT-v2 tokenizer architecture**. T-Causal Conv refers to temporally causal convolution.

A.3    Video Compression Evaluation

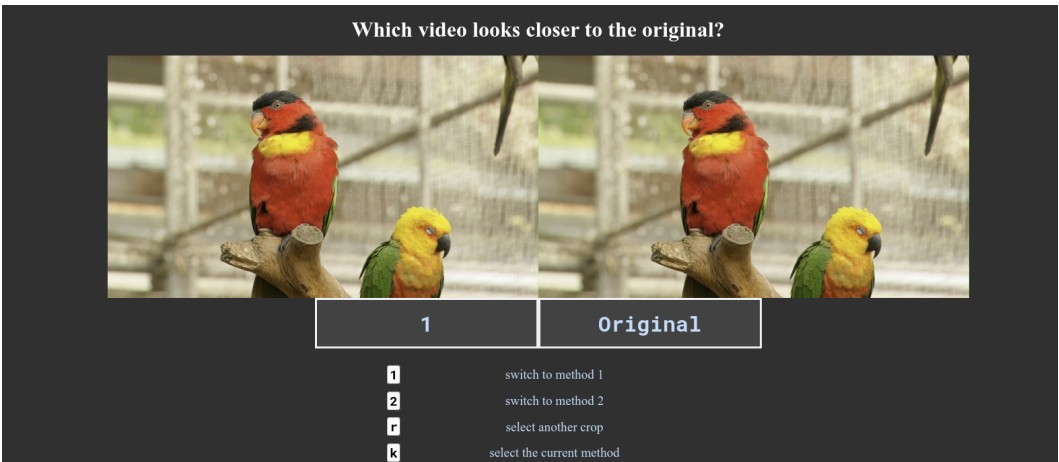

Figure 8: **Rating interface for subjective compression evaluation**.

To rate the quality of the different methods, we use a two-alternative forced choice rating methodology (Fechner, 1860). As this methodology produces a sequence of binary decisions, we calculate Elo scores (Elo & Sloan, 2008) based on pairwise preferences to quantify the relative visual quality between the models. The study was conducted on the 30 videos of the MCL-JCV dataset (Wang et al., 2016), scaled down to a resolution of 640×360 pixels. Sixteen raters are engaged, each providing responses to an average of roughly 800 pairwise-preference questions. The questions are presented with an interface that parallels the one used for the Challenge on Learned Image Compression (http://compression.cc/), extended to comparing videos, as shown in Fig. 8. Raters are instructed to compare the two videos and are not allowed to pause the videos.

A.4    Video Understanding Experiments

**Tokens as prediction targets.** BEiT (Bao et al., 2021) and BEVT (Wang et al., 2022) class of models pretrain visual encoders on pixel inputs by predicting tokens as targets in a masked-modeling framework, and demonstrate state-of-the-art downstream results. We use a simplified BEVT pre-training setup to test the effectiveness of our video tokens as targets for masked modeling. The main difference is that we drop the image-stream from pre-training and only use the video stream and for this reason, we also drop the multiple decoders completely and adopt an encoder-only architecture similar to BEiT. Detailed pre-training and fine-tuning setup is presented in Tab. 6. In Tab. 4 of the main paper, we show that our video tokens are effective targets for masked modeling based video understanding.

**Tokens as inputs.** In Tab. 4, we show that we can re-use video understanding models trained on pixels using our video tokens as input, with very minimal performance drop. For this experiment, we train a factorized variant of the ViViT model (Arnab et al., 2021) on pixels, and evaluate it on de-tokenized pixels from our model. We use the same hyper-parameters as used in Arnab et al. (2021) with a Base sized model operating on 32 frames of inputs at 224p resolution. For the Kinetics-600 experiment, we use the same hyper-parameters as the Kinetics-400 experiments.

B    Additional Results

For better visualization, the generated video samples can be viewed at https://magvit.cs.cmu.edu/v2.

Table 6: **Experimental configurations with tokens as targets**.

| Config | SSv2 Pre-Training | SSv2 Fine-tuning |
|---|---|---|
| inputs | pixels | pixels |
| input size | $16 \times 224 \times 224 \times 3$ | $16 \times 224 \times 224 \times 3$ |
| targets | tokens | classes |
| encoder | ViT-B | ViT-B |
| decoder | linear | linear |
| masking | block-tube (Wang et al., 2022) | none |
| masking ratio | 0.75 | 0.0 |
| mask temporal length | 16 | 0 |
| batch size | 1024 | 512 |
| training epochs | 800 | 50 |
| ViT sequence length | $8 \times 16 \times 16$ | $8 \times 16 \times 16$ |
| **optimization** | | |
| optimizer | AdamW | AdamW |
| optimizer momentum | 0.9 | 0.9 |
| layer decay | 0.75 | 0.75 |
| weight decay | 0.05 | 0.05 |
| learning rate schedule | cosine decay | cosine decay |
| warmup epochs | 40 | 5 |
| **data augmentations** | | |
| random horizontal flip | true | false |
| label smoothing | 0.1 | 0.1 |
| mixup | none | 0.8 |
| cutmix | none | 1.0 |
| droppath | 0.0 | 0.1 |
| dropout | 0.1 | 0.0 |
| random color augmentation | false | false |

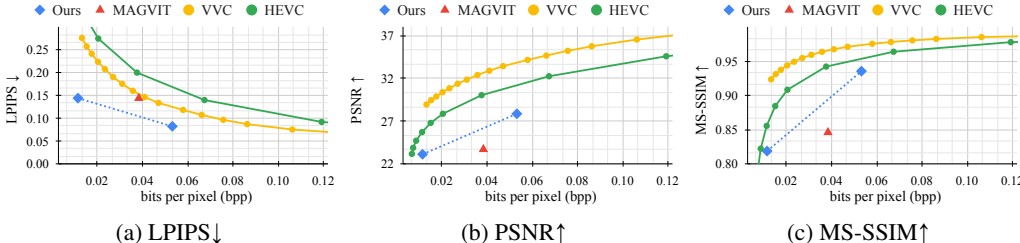

(a) LPIPS↓        (b) PSNR↑        (c) MS-SSIM↑

Figure 9: **Video compression metrics**, supplementary to Tab. 3.

**Where are the text-to-image results?**    We want to emphasize that our goal is to develop a video tokenizer, and many of the proposed techniques are designed specifically for videos. Text-to-image may be out of the scope of our paper. We are currently training text-to-video models that require considerable computational resources. Due to time constraints, these results are not available at the moment. We intend to add the generated videos in the next revision. However, it is important to note that comparing these text-to-image or text-to-video models scientifically is challenging. These models were trained on different datasets, and some were even based on proprietary or non-public data, all under varying training conditions.

Table 7: **Class-conditional image generation on ImageNet 256×256**. Guidance indicates the classifier-free diffusion guidance (Ho & Salimans, 2021). * indicates usage of extra training data. We adopt the evaluation protocol and implementation of ADM.

| Type | Method | w/o guidance | | w/ guidance | | # Params | Steps |
|------|--------|------|------|------|------|----------|-------|
| | | FID↓ | IS↑ | FID↓ | IS↑ | | |
| GAN | BigGAN-deep (Brock et al., 2018) | 6.95 | 171.4 | | | 160M | 1 |
| GAN | StyleGAN-XL (Sauer et al., 2022) | | | 2.30 | 265.1 | 166M | 1 |
| Diff. + VAE* | LDM-4 (Rombach et al., 2022) | 10.56 | 103.5 | 3.60 | 247.7 | 400M | 250 |
| Diff. + VAE* | DiT-XL/2 (Peebles & Xie, 2022) | 9.62 | 121.5 | 2.27 | 278.2 | 675M | 250 |
| Diff. + BAE | Binary latent diffusion (Wang et al., 2023) | 8.21 | 162.3 | | | 172M | 64 |
| Diffusion | ADM+Upsample (Dhariwal & Nichol, 2021) | 7.49 | 127.5 | 3.94 | 215.8 | 608M | 2000 |
| Diff. + VAE* | MDT (Gao et al., 2023) | 6.23 | 143.0 | 1.79 | 283.0 | 676M | 250 |
| Diff. + VAE* | MaskDiT (Zheng et al., 2023) | 5.69 | 178.0 | 2.28 | 276.6 | 736M | 40 |
| Diffusion | CDM (Ho et al., 2022b) | 4.88 | 158.7 | | | | 8100 |
| Diffusion | RIN (Jabri et al., 2023) | 3.42 | 182.0 | | | 410M | 1000 |
| Diffusion | simple diffusion (Hoogeboom et al., 2023) | 2.77 | 211.8 | 2.44 | 256.3 | 2B | 512 |
| Diffusion | VDM++ (Kingma & Gao, 2023) | 2.40 | 225.3 | 2.12 | 267.7 | 2B | 512 |
| AR-LM + VQ | VQGAN (Esser et al., 2021) | 15.78 | 78.3 | | | 1.4B | 256 |
| MLM + VQ | MaskGIT (Chang et al., 2022) | 6.18 | 182.1 | | | 227M | 8 |
| MLM + VQ | Token-Critic (Lezama et al., 2022) | 4.69 | 174.5 | | | 368M | 36 |
| MLM + VQ | Contextual RQ-Transformer (Lee et al., 2022) | 3.41 | 224.6 | | | 1.4B | 72 |
| MLM + VQ | DPC (Lezama et al., 2023) | 4.45 | 244.8 | | | 454M | 180 |
| MLM + LFQ | *MAGVIT-v2 (this paper)* | 3.65 | 200.5 | **1.78** | **319.4** | 307M | 64 |

Table 8: **Video generation results**: class-conditional generation on UCF-101 with AR-LM models. We use the same transformer configuration as MLM experiments but without vocabulary factorization and weight tying. As a result, the AR-LM with MAGVIT-v2 uses more parameters in the embedding table and the softmax layer.

| Tokenizer | FVD↓ | #Params | #Steps |
|-----------|------|---------|--------|
| MAGVIT (Yu et al., 2023a) | 265 | 306M | 1024 |
| *MAGVIT-v2 (this paper)* | **109** | 840M | 1280 |

