# OpenReview forum: "Language Model Beats Diffusion - Tokenizer is key to visual generation"
_ICLR.cc/2024/Conference — ICLR 2024 poster_

### Official Review · Reviewer_k12R · 2023-10-30

**Soundness:** 3 good
**Presentation:** 3 good
**Contribution:** 4 excellent
**Rating:** 8
**Confidence:** 4

**Summary:**

This paper introduces a novel visual tokenizer designed to enhance large language models in producing high-quality images and videos. Experimental results show that, when integrated with the proposed tokenizer, LLM surpasses diffusion models in standard benchmarks such as ImageNet, UCF-101, and Kinetics-600. Additionally, the paper presents promising results in video compression and representation learning.

**Strengths:**

1.	This paper presents the first evidence of large language models surpassing diffusion models on the ImageNet benchmark.
2.	The paper proposes a novel lookup-free quantization approach, providing a promising direction for expanding vocabulary size in LLM-based visual generation.
3.	The motivation is clear, and the overall presentation is coherent and easy to follow.
4.	Good results on visual generation video compression, and video representation learning.

**Weaknesses:**

1.	While the presented method is tailored for masked LM, many of the prevailing and powerful LLMs, such as LLaMA [A], employ an autoregressive approach. Incorporating results from AR-LM would greatly enhance the paper's relevance to the community.
2.	In Table 4, despite the good action recognition performance showcased by the proposed method, it doesn't conclusively establish its efficacy as a viable self-supervised pre-training target. Notably, some pivotal baselines, like pixel colors and the image descriptor from MaskFeat [B], are missing.

[A] Touvron, H., Lavril, T., Izacard, G., Martinet, X., Lachaux, M. A., Lacroix, T., ... & Lample, G. Llama: Open and efficient foundation language models. arXiv preprint arXiv:2302.13971.

[B] Wei, C., Fan, H., Xie, S., Wu, C. Y., Yuille, A., & Feichtenhofer, C. Masked feature prediction for self-supervised visual pre-training. In CVPR 2022.

**Questions:**

1.	In Figure 1, it's highlighted that the VQ generation FID sees a pivotal change at a vocabulary size of 2^14, while the LFG generation FID consistently improves. I'm curious to understand how the LFG generation FID would respond to even larger vocabulary sizes.
2.	Regarding Table 3, what could be the reason behind the proposed method's PSNR and MS-SSIM values being inferior to those of the standard video codec?

---

> ### Author Response · Authors · 2023-11-23
> **Response to Reviewer k12R**
>
> We appreciate the thoughtful comments from reviewer k12R and the acknowledgment that `with the proposed tokenizer, LLM surpasses diffusion models in standard benchmarks`, and the tokenizer also shows `promising results in video compression and representation learning.`
>
> > W1: Incorporating results from AR-LM.
>
> In the Appendix, we have added experiments on the autoregressive language model (AR-LM) where the proposed tokenizer outperforms the prior work MAGVIT [R3-A]. The results, also provided in Table R3.1 below, show our tokenizer can also work with the AR-LM.
>
> It is worth noting that we have observed that achieving optimal performance requires a much larger AR-LM model size. But due to time constraints, we have not been able to train a larger AR-LM model with an adequate number of training steps. That also partly explains our current choice of not building our work on the autoregressive language model, and we leave it as future work.
>
> *Table R3.1: Video generation results: class-conditional generation on UCF-101 with AR-LM models. We use the same transformer configuration as MLM experiments but without vocabulary factorization and weight tying. As a result, the AR-LM with our tokenizer uses more parameters in the embedding table and the softmax layer.*
>
> | Tokenizer | FVD↓ | # Params | # Steps |
> |---|:---:|:---:|:---:|
> | MAGVIT | 265 | 306M | 1024 |
> | *This paper* | **109** | 840M | 1280 |
>
> > > [R3-A] [MAGVIT: Masked Generative Video Transformer. In CVPR 2023.](http://openaccess.thecvf.com/content/CVPR2023/html/Yu_MAGVIT_Masked_Generative_Video_Transformer_CVPR_2023_paper.html)
>
> > W2: Efficacy as a viable self-supervised pre-training target compared to baselines like pixel colors and image descriptors.
>
> Following the reviewer’s suggestion, we have added pixel colors and HoG descriptors from MaskFeat [B] as baseline self-supervised pre-training targets into revised Table 4, with a distilled version in Table R3.2 below. As shown, pre-training with tokens from our model as the target achieves the best performance compared to the previous leading tokenizer MAGVIT as well as other baselines, including raw pixels, 3D VQ-VAE, and HoG descriptors.
>
> *Table R3.2 Video action recognition performance (classification accuracy↑ $\times$100) with different self-supervised pre-training targets as the transformer output.*
>
> | Target | SSv2 |
> |---|:---:|
> | 3D VQ-VAE | 64.13 |
> | MAGVIT | 67.22 |
> | *This paper* | **67.38** |
> | Raw pixel | 64.83 |
> | HoG descriptor | 65.86 |
>
> > Q1: Generation performance with LFQ at larger vocabularies?
>
> In Table R3.3, we showcase an instantiation of LFQ with a much larger vocabulary size at $2^{40}$. As shown, it yields consistent improvement over $2^{18}$ in both reconstruction and generation FID metrics.
>
> *Table R3.3 Class-conditional image generation results on ImageNet 512$\times$512 with larger LFQ vocabularies. We use an MLM model and 12 decoding steps.*
>
> | Vocabulary | Groups | # Params | rFID↓ | FID↓ |
> |:---:|:---:|:---:|:---:|:---:|
> | $2^{18}$ | 2 | 307M | 1.22 | 5.93 |
> | $2^{40}$ | 4 | 312M | **0.80** | **5.15** |
>
>
> > Q2: PSNR and MS-SSIM compared to standard video codec.
>
> Our model uses GAN and perceptual losses to improve the reconstruction quality especially in terms of the realism of the generated video. At a similarly low bit rate, standard codecs based on block-level Fourier-related transformations may preserve better local details but introduce inter-block artifacts. As pointed out by [R3-B], traditional distortion metrics such as PSNR and MS-SSIM, “for very low bitrates … these distortion metrics **lose significance** as they favor pixel-wise preservation of local structure over preserving texture and global structure.” [R3-B]
> As a result, while standard codecs might appear to outperform on metrics like PSNR and MS-SSIM, they fall short in the gold-standard human rater study (see Figure 6). We have revised Section 4.3 to clarify this point.
>
> > > [R3-B] [Generative adversarial networks for extreme learned image compression](http://openaccess.thecvf.com/content_ICCV_2019/html/Agustsson_Generative_Adversarial_Networks_for_Extreme_Learned_Image_Compression_ICCV_2019_paper.html)

---

### Official Review · Reviewer_sfJe · 2023-10-31

**Soundness:** 3 good
**Presentation:** 3 good
**Contribution:** 4 excellent
**Rating:** 8
**Confidence:** 4

**Summary:**

The paper focuses on learning a video / image tokenizer to discretize video / images so that they can be modeled using a Language Model. They specifically introduce one innovation in this setting: A lookup free quantizer. They show that in this limit of using a large vocabulary and no lookup, the tokenizer reconstruction and LM generation quality both increase with vocabulary size. The authors also show that the learned tokenizer performs very well as a compression scheme.

**Strengths:**

The paper does a good job of motivating the core thesis of the paper: How to design a tokenizer for image / video? The authors also do a good job of presenting this idea to the uninitiated. It is also pretty clear that the work is significant to academia and industry given that it helps unify image generation with image understanding and natural language generation and understanding techniques. The ideas in the paper are well explained. The authors also do a thorough job of running experiments to substantiate many claims including numerous ablations. Some of the important technical insights like the lookup free quantizer (and in general lower dimensional code words) helping in generation quality are substantiated by experimental results

**Weaknesses:**

My main concern is the completeness of the exposition in the paper. The authors assume that the reader is familiar with the state of the art in video tokenization and details do get rather buried in the many “deltas” relative to the baseline. I do understand the space limitations but it might be helpful if the authors try to make the core system / model design more explicit. Lot of the ideas like factorization of the output space in the decoder (and associated weight tying) for example are just mentioned in passing.

**Questions:**

* The question “Why masked LM and not AR LM for image / video generation?” for evaluating the tokenizer was not clearly answered.
* No explicit definition of the objective for training the tokenizer (loss function)
* No mention of decoder in VQ-VAE and VQ-VAE loss used when we use no lookup
* More motivation needed on why the authors choose to use a causal encoder for tokenizer when doing masked LM for image / video generation
* It’s not clear why the authors tackle video generation if the aim was to understand the fundamentals of tokenization. It may be desirable for them to clearly motivate why they study videos and not images alone?
* It may be interesting for the reader to understand the computational complexity of both the tokenizer (encoder) and the detokenizer (decoder) and how they compare with video or audio codecs

---

> ### Author Response · Authors · 2023-11-23
> **Response to Reviewer sfJe**
>
> We appreciate the constructive comments from reviewer sfJe and the positive assessment. We provide responses to each individual question below.
>
> > W1: the completeness of exposition in the paper; assuming the reader is familiar with the state of the art in video tokenization.
>
> Thank you for your understanding regarding the space limitations. To enhance the method's readability and self-containment, we have implemented the following changes. First, we have added a paragraph in Section 3.1, detailing all the training losses, which includes those from prior works. Second, we have added a section to the Appendix, which provides additional details on model design and hyperparameters. Third, a paragraph has been included in Section 3.2 to clarify the weight tying approach in the factorized prediction. We hope these can improve the completeness of the exposition, and we are open to additional recommendations from the reviewer.
>
> > Q1: Why masked LM and not AR LM for image / video generation?
>
> We selected the masked language model (MLM) due to its competitive performance on benchmark datasets [R2-A, R2-B] compared to the autoregressive language model (AR-LM), where MLM also uses much fewer decoding steps than AR-LM. In Table R2.1, we also show experiments on the AR-LM where the proposed tokenizer outperforms the prior work MAGVIT [R2-A]. The results show our tokenizer can also work with the AR-LM, but the AR-LM lags behind the state-of-the-art. We have included this rationale in the revised Section 4.2 and added the new result to the Appendix.
>
> It is worth noting that we have observed that achieving optimal performance requires a much larger AR-LM model size. But due to time constraints, we have not been able to train a larger AR-LM model with an adequate number of training steps. That also partly explains our current choice not to build our work on the autoregressive language model. We are investigating the integration of the tokenizer with the large AR-LM and will study this in our future research.
>
> *Table R2.1: Video generation results: class-conditional generation on UCF-101 with AR-LM models. We use the same transformer configuration as MLM experiments but without vocabulary factorization and weight tying. As a result, the AR-LM with our tokenizer uses more parameters in the embedding table and the softmax layer.*
>
> | Tokenizer | FVD↓ | # Params | # Steps |
> |---|:---:|:---:|:---:|
> | MAGVIT | 265 | 306M | 1024 |
> | *This paper* | **109** | 840M | 1280 |
>
> > > [R2-A] [MAGVIT: Masked Generative Video Transformer. In CVPR 2023.](http://openaccess.thecvf.com/content/CVPR2023/html/Yu_MAGVIT_Masked_Generative_Video_Transformer_CVPR_2023_paper.html)
> > > [R2-B] [Discrete Predictor-Corrector Diffusion Models for Image Synthesis. In ICLR 2023.](https://openreview.net/forum?id=VM8batVBWvg)
>
>
> > Q2: No explicit definition of the objective for training the tokenizer (loss function).
>
> Following the reviewer's recommendation, we have added a description of the tokenizer's training objective at the end of Section 3.1, detailing all the training losses including those from prior works. In addition to the entropy penalty in Equation 5, the overall training objective involves the standard combination of reconstruction, GAN, perceptual, and commitment losses from VQGAN [R2-C], excluding the codebook loss. In addition, we follow [R2-A] in using LeCAM regularization [R2-D] for improved stability. To provide better clarity on the details, we have added Appendix A.2 listing all relevant hyperparameters.
>
> > > [R2-C] [Taming Transformers for High-Resolution Image Synthesis. In CVPR 2021.](https://openaccess.thecvf.com/content/CVPR2021/html/Esser_Taming_Transformers_for_High-Resolution_Image_Synthesis_CVPR_2021_paper.html)
> > > [R2-D] [Regularizing Generative Adversarial Networks under Limited Data. In CVPR 2021.](http://openaccess.thecvf.com/content/CVPR2021/html/Tseng_Regularizing_Generative_Adversarial_Networks_Under_Limited_Data_CVPR_2021_paper.html)
>
> > Q3 No mention of decoder in VQ-VAE and VQ-VAE loss used when we use no lookup.
>
> Initially, the decoder was only briefly mentioned in Section 2 and in the caption of Figure 2. We have improved the clarity by elaborating on the additional details on model design and hyperparameters in Appendix A.2 with an architecture diagram in Figure 7, as well as by providing a discussion of the loss functions in Section 3.1.

---

> > ### Author Response · Authors · 2023-11-23
> > **Response to Reviewer sfJe (Cont.)**
> >
> > >Q4: Motivation why the authors choose to use a causal encoder for tokenizer when doing masked LM for image / video generation?
> >
> > We have found that causal encoders facilitate frame prediction and result in improved generation FVD, as shown in Table R2.2. We have added this comparison into the revised Table 1 and Section 4.2. Additionally, this approach enables the representation of image and video within a unified token space, which could assist the community to explore joint image-video training.
> >
> > Nevertheless, we agree with the reviewer that the causal design is also suitable for autoregressive language modeling. We have included a preliminary study of this in the Appendix and also in Table R 2.1. The result shows that our tokenizer also provides benefits for the autoregressive language model. Due to the reasons outlined in Q1, we chose to focus on using the masked LM as the backbone for generation.
> >
> > *Table R2.2: Comparison of non-causal and causal tokenizer on the Kinetics-600 frame prediction task. A masked LM with 307M parameters is employed using 12 inference steps.*
> >
> > | Causal | K600 FVD↓ |
> > |:---:|:---:|
> > | No | 11.6±0.6 |
> > | *Yes* | **5.2±0.2** |
> >
> >
> > > Q5: Why tackle video generation? Motivate why studying videos and not images alone?
> >
> > Existing generation models, such as the stable diffusion model, have achieved remarkable results in image generation. Compared to image generation, video generation still faces substantial challenges in generating consistent and realistic motion. We are interested in exploring the capabilities of language models in tackling this unsolved challenge. Therefore, this paper focuses on the video tokenizer that can effectively model spatial-temporal dynamics for generative modeling. We have clarified this point in the beginning of Section 3 of the revised manuscript.
> >
> > > Q6: Computational complexity of tokenizer and detokenizer? Comparison with codecs?
> >
> > Following the reviewer’s recommendation, we evaluate the computational complexity of our models and MAGVIT using MFLOPs (megaFLOPs) per pixel and frames per second, as shown in Table R2.3. At a similar complexity, our model achieves a higher compression ratio (lower bpp) while maintaining a similar quality (LPIPS) compared to MAGVIT [R2-A]. With a higher complexity, our model demonstrates significantly higher quality as measured by LPIPS and also human rater study shown in Figure 6.
> >
> > We find it difficult to compare our method with the standard codecs because our model currently runs on accelerators with floating point operations while standard codecs are often implemented on CPU using integer operators or supported by hardware encoder/decoders. We leave this comparison to our future work.
> >
> > We acknowledge that, judging from its network size and FLOPs, our current model may incur higher computational costs using TPUs compared to the common setups of standard codecs, e.g. HEVC Main Profile. We have added a short discussion in the revised Section 4.3. However, inspired by the impressive advancements in the acceleration of neural networks, we are cautiously optimistic about future research in this direction that could significantly improve the efficiency of the tokenizer.
> >
> > *Table R2.3: Computational complexity of tokenizers at 360p resolution (480x360). Frames per second is measured with a single TPUv4 chip.*
> >
> > | Model | bpp | LPIPS↓ | MFLOPs per pixel↓ || Frames per second↑ ||
> > |---|:---:|:---:|:---:|:---:|:---:|:---:|
> > |  |  |  | Tokenizer | Detokenizer | Tokenizer | Detokenizer |
> > | MAGVIT | 0.0384 | 0.1440 | 5.6 | 10.8 | 133 | 50 |
> > | *This paper* | **0.0147** | 0.1442 | 6.1 | 8.0 | 131 | 67 |
> > | *This paper* | 0.0532 | **0.0823** | 15.9 | 21.3 | 54 | 29 |

---

### Official Review · Reviewer_rVyC · 2023-11-04

**Soundness:** 3 good
**Presentation:** 3 good
**Contribution:** 4 excellent
**Rating:** 8
**Confidence:** 3

**Summary:**

The paper proposes a novel visual tokenizer based on lookup-free quantization (LFQ). With the growth of the vocabulary size LFQ consistently improve both reconstruction and generation quality, which is in stark contrast with Vector Quantization (VQ) where an increased vocabulary size reduces reconstruction error but hurts generation results. The tokenizer can be integrated with MAGVIT and achieves state-of-the-art performance on video generation. The tokenizer can also improve video compression and video recognition.

**Strengths:**

1. Starting from an interesting finding that enlarging the vocabulary improves reconstruction quality but hurts generation results, the paper proposes solutions (LFQ) to tame both reconstruction and generation simultaneously.

2. A detailed study of architecture modifications that improves up MAGVIT supported by extensive ablations.

3. The tokenizer is proved to benefit video generation, compression and recognition. It will potentially have a huge impact on the general audience of video understanding.

**Weaknesses:**

1. For video compression results, it would be better if there is a PSNR/LPIPS/MS-SSIM-bpp curve comparing the performance across different bpps.

2. In the video recognition setup, it seems unnecessary to detokenize the visual tokens back to pixels since BEVT and BEIT can work with tokenized input. I understand one of the main reasons is that the underlying recognition model is the ViViT which takes raw pixels as input (as stated in the draft). However, you may also have a comparison with BEVT.

**Questions:**

See weakness.

---

> ### Author Response · Authors · 2023-11-23
> **Response to Reviewer rVyC**
>
> We appreciate the valuable feedback from Review rVyC and the acknowledgment of our method that `is proved to benefit video generation, compression and recognition` and `potentially has a huge impact on the general audience of video understanding`. Please see our response below.
>
> > W1: Video compression metric curves.
>
> Following the suggestion, we have added Figure 9 in the Appendix showing the curves of compression metrics, including LPIPS, PSNR, and MS-SSIM at various bpp levels. The results are consistent with the findings from Table 3 that `our model outperforms MAGVIT on all metrics and outperforms all methods on LPIPS`. It is worth noting that LPIPS may `correlate more closely with subjective assessments`, which we have shown in the human rater study in Figure 6, than PSNR or MS-SSIM, according to [R1-A].
>
> > > [R1-A] [Generative adversarial networks for extreme learned image compression](http://openaccess.thecvf.com/content_ICCV_2019/html/Agustsson_Generative_Adversarial_Networks_for_Extreme_Learned_Image_Compression_ICCV_2019_paper.html)
>
>
> > W2: Necessity of detokenization in video recognition.
>
> We appreciate your understanding that ViViT `takes raw pixels as input`. The detokenization experiments used the **frozen** ViViT model that was trained on RGB pixels. For proper inference, the detokenizer is needed to convert the tokens back into pixel inputs, aligning with the input requirements of the frozen ViViT model. As shown in Table 4, our model shows superior performance compared to MAGVIT [R1-B] and is very close to raw pixels while only using ~1/500 data bandwidth. This setting was mentioned in Appendix A.3 of the original submission. And we have revised Section 4.4 to clarify this.
>
> Following the reviewer’s suggestion, we also train a BEVT model with tokens as both the input and the output for action recognition. The result is shown in Table R1.1. We find that directly using token inputs yields suboptimal performance in part because the current architecture and pre-training setups are tailored for pixel inputs. Further research is needed to investigate this token-in-token-out pretraining paradigm for action recognition, where both the input and output are based on discrete tokens.
>
> *Table R1.1: Video action recognition performance (classification accuracy↑ $\times$100) on SSv2 dataset.*
>
> | Tokenizer | Input+Output | Output only |
> |---|:---:|:---:|
> | MAGVIT | 60.86 | 67.22 |
> | *This paper* | **60.99** | **67.38** |
>
> > > [R1-B] [MAGVIT: Masked Generative Video Transformer. In CVPR 2023.](http://openaccess.thecvf.com/content/CVPR2023/html/Yu_MAGVIT_Masked_Generative_Video_Transformer_CVPR_2023_paper.html)

---

### Meta-Review · Area_Chair_RCsB · 2023-12-09

**Metareview:**

This paper introduces a novel approach in the form of a visual tokenizer based on lookup-free quantization (LFQ). The paper has been rigorously reviewed by three experts in the field. The reviewers collectively appreciate the significant contribution of this paper, notably presenting the first evidence that large language models can surpass diffusion models in the ImageNet benchmark. This accomplishment is a noteworthy advancement in the field of generative modeling.

Reflecting on the constructive feedback provided by the reviewers, I am pleased to recommend this paper for acceptance at ICLR 2024. This decision underscores the paper's innovative nature and its potential impact on the research community. The authors are encouraged to carefully address remaining issues raised by reviewers in the final camera-ready version of the manuscript. Congratulations again to the authors on the acceptance of their paper.

**Justification For Why Not Higher Score:**

The extensiveness of the experiments on video recognition and compression could be improved by incorporating more baselines from relevant tracks.

**Justification For Why Not Lower Score:**

This work has an important and interesting finding that enlarging the vocabulary improves reconstruction quality but hurts generation results.

---

### Decision · Program_Chairs · 2024-01-16

Accept (poster)